# PLK-1 Interacting Checkpoint Helicase, PICH, Mediates Cellular Oxidative Stress Response

**Anindita Dutta, Apurba Das** , **Deepa Bisht** , **Vijendra Arya and Rohini Muthuswami ***

Chromatin Remodeling Laboratory, School of Life Sciences, Jawaharlal Nehru University,
New Delhi 110067, India
* Correspondence: rohini_m@mail.jnu.ac.in

**Abstract:** Cells respond to oxidative stress by elevating the levels of antioxidants, signaling, and transcriptional regulation, often implemented by chromatin remodeling proteins. The study presented here shows that the expression of PICH, a Rad54-like helicase belonging to the ATP-dependent chromatin remodeling protein family, is upregulated during oxidative stress in HeLa cells. We also show that PICH regulates the expression of Nrf2, a transcription factor regulating antioxidant response in both the absence and presence of oxidative stress. The overexpression of *PICH* in *PICH*-depleted cells restored *Nrf2* as well as antioxidant gene expression. In turn, Nrf2 regulated the expression of *PICH* in the presence of oxidative stress. ChIP experiments showed that PICH is present on the *Nrf2* as well as antioxidant gene promoters, suggesting that the protein might be regulating the expression of these genes directly by binding to the DNA sequences. In addition, Nrf2 and histone acetylation (H3K27ac) also played a role in activating transcription in the presence of oxidative stress. Both Nrf2 and H3K27ac were found to be present on *PICH* and antioxidant promoters. Their occupancy was dependent on the *PICH* expression as fold enrichment was found to be decreased in *PICH*-depleted cells. PICH ablation led to the reduced expression of Nrf2 and impaired antioxidant response, leading to increased ROS content and thus showing PICH is essential for the cell to respond to oxidative stress.

**Keywords:** PICH; Nrf2; oxidative stress; epigenetics; ATP-dependent chromatin remodeling; antioxidants; $H_2O_2$



## 1. Introduction

The ATP-dependent chromatin remodeling proteins dynamically alter the chromatin architecture to enable transcription factors to access the genomic DNA and thereby mediate transcription and repair [1]. Organismal life encounters reactive oxidants from internal metabolism as well as from environmental toxicant exposure, resulting in oxidative stress due to an imbalance between reactive oxygen species (ROS) and antioxidants. The role of epigenetic modulators in combating oxidative stress has been well documented. For example, oxidative stress has been shown to activate histone acetyltransferases and inhibit histone deacetylase activity [2]. Studies have also shown that BRG1, an ATP-dependent chromatin remodeling protein, interacts with Nrf2, a transcription factor, to regulate the expression of *HO-1* in response to oxidative stress [3]. CHD6 too is an important regulator of oxidative stress [4].

PICH (PLK-1 interacting checkpoint helicase), also known as ERCC6L, is a Rad54-like helicase belonging to the ATP-dependent chromatin remodeling protein family [5–7]; however, it does not have any chromatin remodeling activity. The protein has been identified as a strong binding partner and substrate of PLK-1 that localizes at the kinetochores [5]. Immunofluorescence staining has revealed that PICH is mostly concentrated between kinetochores in prometaphase cells, while in metaphase cells, the protein localizes to numerous short threads that stretch between the sister kinetochores of the aligned chromosomes [5].

The protein has been shown to respond to the tension-dependent alterations in DNA topology by resolving the ultrafine bridges generated between the centromeres of the segregating chromatids and regulating the spindle attachment checkpoint [8]. PICH-depleted cells show chromosomal abnormalities, dead cells, bi-nucleus, and multi-nucleus formation [9]. The protein is found to be overexpressed in many cancers, and silencing the expression of PICH has been shown to inhibit their proliferation [10,11]. Studies have also shown that the protein interacts with BEND3 as well as with topoisomerases; however, the protein has not been shown to reposition/remodel nucleosomes [9,12,13].

Previously, we showed that the expression of BRG1 and SMARCAL1, two ATP-dependent chromatin remodelers, was upregulated at both the transcription and protein levels in doxorubicin-treated HeLa cells [14]. We showed that this upregulation was needed for the activation of the DNA damage response pathway [15,16]. To understand whether this upregulation was needed during oxidative stress, we performed a preliminary experiment wherein we transfected HeLa cells with the D-amino acid oxidase gene. On treatment with D-serine, $H_2O_2$ was produced inside the cells, resulting in oxidative stress. We then analyzed the expression of ATP-dependent remodelers *BRG1*, *SMARCAL1*, *RAD54L*, *ZRANB3*, and *INO80*, which are known to play a role in DNA damage response/repair [14,17–19]. In this experiment, we used PICH as a negative control as reports were that this protein cannot remodel nucleosomes [9]. Further, there are no reports that PICH plays a role in DNA damage response/repair. To our surprise, we found that *BRG1*, *SMARCAL1*, *RAD54L*, *ZRANB3*, and *INO80* were unchanged but *PICH* was upregulated. This led us to hypothesize that maybe PICH has a role to play in oxidative stress, either in the DNA damage response pathway or in modulating the expression of antioxidant genes as well as of Nrf2.

The nuclear factor erythroid 2 (NFE2)-related factor 2 (Nrf2) is a member of the cap 'n'collar (CNC) subfamily of basic region leucine zipper transcription factors [20,21]. Studies over the past decade have established the role of Nrf2 in combating oxidative stress in cells [22]. In cells, in the absence of oxidative stress, Nrf2is ubiquitinated by Keap1 and degraded by the proteasome pathway [23,24]. In the presence of oxidative stress, Keap1 is inactivated, leading to the release of Nrf2, which migrates to the nucleus and activates the expression of antioxidant genes by binding to the antioxidant responsive element (ARE) present on the promoters of the antioxidant genes [23,24]. Knockout of *Nrf2* in mice increases their susceptibility to chemical toxins, and the mice exhibit disease conditions associated with oxidative pathology [25].

In this paper, the role of PICH during oxidative stress has been investigated. We provide evidence that PICH regulates the antioxidant response in HeLa cells under oxidative stress. PICH also regulates the expression of *Nrf2* and together with Nrf2 and H3K27ac appears to drive the expression of the antioxidant genes.

## 2. Results

### 2.1. PICH Expression Is Upregulated When Cells Are Exposed to Oxidative Stress

Previously, we reported that BRG1 and SMARCAL1 both belonging to the ATP-dependent chromatin remodeling protein family co-regulate each other's expression on treatment with doxorubicin [14]. To investigate whether this regulation is universal and occurs on the induction of any type of DNA damage, HeLa cells were subjected to oxidative stress generated endogenously by transfecting cells with a plasmid encoding for D-amino acid oxidase (DAAO), which catalyzes the oxidative deamination of D-amino acids, including D-serine, to generate keto acids, ammonia, and hydrogen peroxide ($H_2O_2$), thus generating oxidative stress endogenously [26].

The cells, 36 h post-transfection, were treated with 25 mM D-serine for 10 min [26], resulting in the production of $H_2O_2$ within the cell by the action of DAAO on D-serine. ROS production was confirmed by DCFDA staining (Figure S1A,B). The treated cells were harvested, and the RNA was isolated to analyze the expression of *BRG1* and *SMARCAL1* using qRT-PCR. Surprisingly, the expression of *BRG1* and *SMARCAL1* was unchanged, but the expression of *PICH*, an ATP-dependent chromatin remodeler, and *Nrf2*, a transcription

factor, was upregulated (Figure S1C). Western blot showed that the protein levels of both PICH and Nrf2 were upregulated (Figure S1D; quantitation provided in Figure S6A). Further, the expression of the antioxidant genes *CAT, GPX1, GSR,* and *TXNRD1* was found to be upregulated, while SOD1 was unchanged (Figure S1E). Concomitantly, catalase activity was also found to be upregulated (Figure S1F). Untransfected HeLa cells treated with D-serine did not exhibit these changes in the transcript, indicating that this molecule by itself is not contributing to oxidative stress or changes thereof (Figure S1G). Taken together, the experimental results indicated that PICH expression was upregulated on oxidative stress.

To confirm the upregulation of PICH in response to oxidative stress, HeLa cells were treated with 100 μM $H_2O_2$, and the expression was analyzed as a function of time [27]. In this experiment, we did not transfect cells with the DAAO vector. Instead, cells were treated with hydrogen peroxide. Analysis showed that the expression of *PICH* was upregulated 20 min post-treatment in comparison with the untreated HeLa cells (Figure 1A). Protein levels were analyzed by Western blot and were also found to be upregulated (Figure 1B; quantitation provided in Figure S6B). It needs to be noted that the protein levels remained upregulated steadily post-$H_2O_2$ treatment while the RNA levels appeared to fluctuate. It is quite possible that the expression of PICH is regulated by both transcriptional and post-transcriptional mechanisms, ensuring that protein levels are steady in the cell even if the transcript levels fluctuate. As the first peak of *PICH* expression was at 20 min post-treatment as compared with the untreated HeLa cells, all the experiments described henceforth were performed after treating HeLa cells with 100 μM $H_2O_2$ for 20 min. Under this condition, the production of ROS was confirmed by staining the cells with DCFDA (Figure 1C,D).

Next, the expression of *Nrf2, BRG1*, and *SMARCAL1* was analyzed after treatment with 100 μM $H_2O_2$ for 20 min. As expected, *Nrf2* expression was upregulated on the exogenous generation of oxidative stress (Figure 1E). Interestingly, unlike the endogenous pathway, on exogenous treatment with $H_2O_2$, *BRG1* and *SMARCAL1* were upregulated along with *Nrf2* (Figure 1E). The protein levels were also found to be upregulated (Figure 1F; quantitation provided in Figure S6C). The dichotomy between the endogenous and exogenous pathways was further accentuated on the analysis of the expression of antioxidant genes. Unlike the endogenous pathway, where *SOD1* was unchanged and *GSR* was upregulated, in the exogenous pathway, *SOD1* was upregulated while *GSR* was unchanged (compare Figure 1G with Figure S1E). As it is not possible to compare the amount of ROS produced under these two conditions, a direct comparison between the amount of ROS produced and the pathway activated cannot be made. However, we noted that catalase activity was upregulated when cells were treated with D-serine after *DAAO* transfection as well as with hydrogen peroxide (Figures 1H and S1F).

*2.2. PICH Regulates the Expression of Nrf2 in HeLa Cells in the Absence and Presence of Oxidative Stress*

To understand whether PICH is necessary for the upregulation of *Nrf2* on oxidative stress, shRNA against the 3′ UTR (Sh*PICH)* as well as CDS (Sh*PICH*_CDS; data shown in Figure S3A–C; quantification of the Western blots shown in S6D) of PICH was used to downregulate the expression of this gene in HeLa cells. The downregulation of *PICH* with these shRNAs led to the downregulation of *Nrf2* in both the absence and presence of $H_2O_2$, indicating that PICH regulates *Nrf2* expression regardless of oxidative stress (Figures 2A and S2A). Western blots showed that the protein levels were also downregulated (Figure 2B and Figure S2B; quantitation of the Western blots provided in Figure S6E,F). Further, the expression of antioxidant genes was also downregulated, as was the catalase activity in both the absence and presence of oxidative stress (Figures 2C,D and S2C,D). We further confirmed that the downregulation was specific by analyzing the expression of *DNA-PKc* and found it to be unchanged in both Sh*PICH* and Sh*PICH*_CDS cells in both the absence and in presence of treatment (Figure S3D).

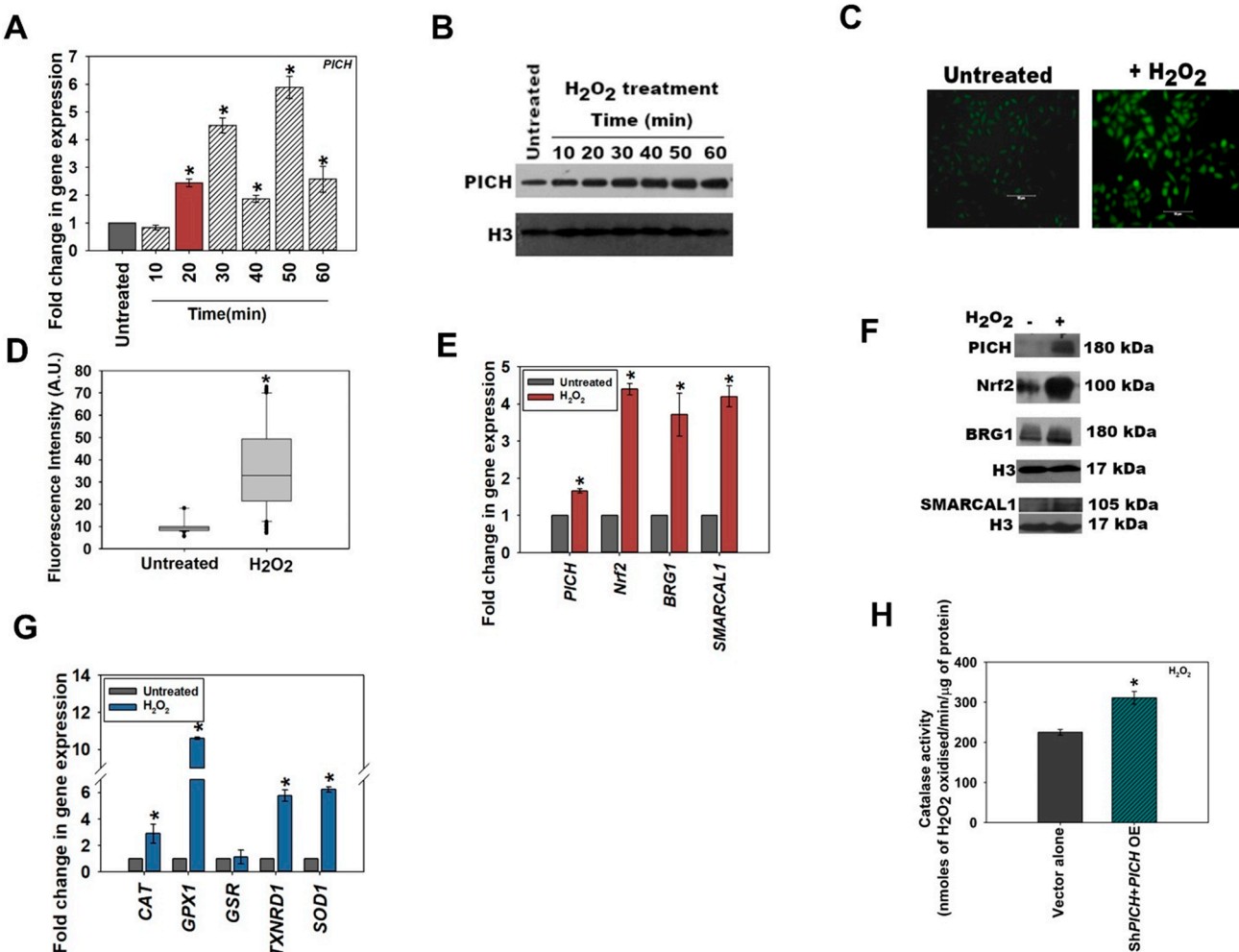

**Figure 1.** PICH expression is upregulated when cells are exposed to oxidative stress. (**A**) The expression of *PICH* was analyzed at indicated time points in HeLa cells after treatment with 100 μM $H_2O_2$ on qRT-PCR. (**B**) The expression of PICH was examined with Western blot. H3 was used as an internal control. (**C**) Cellular ROS was analyzed using DCFDA in HeLa cells in the absence and presence of $H_2O_2$. (**D**). Fluorescent intensity was quantitated using the software provided by TiE, Nikon Microscope. (**E**) The expression of *PICH*, *Nrf2, BRG1*, and *SMARCAL1* were quantitated using qRT-PCR. (**F**) The expression of PICH, Nrf2, BRG1, and SMARCAL1 was analyzed with western blot. (**G**) The expression of antioxidant genes *CAT*, *GPX1*, *GSR*, *TXNRD1,* and *SOD1* WAS analyzed using qRT-PCR. (**H**) Catalase activity (μmol/min) was quantitated in untreated and treated (100 mM $H_2O_2$; 20 min) HeLa cells. *GAPDH* was used as the internal control in all the qRT-PCR experiments. The qRT-PCR experiments are presented as average ± SEM of three independent (biological) experiments with each biological experiment performed as two technical replicates. A star (asterisk) indicates that *p*-value is < 0.05. The Western blots were repeated twice (biological independent experiments), and the best blot is represented in the figure. The catalase activity is presented as the average ± SEM of three independent experiments.

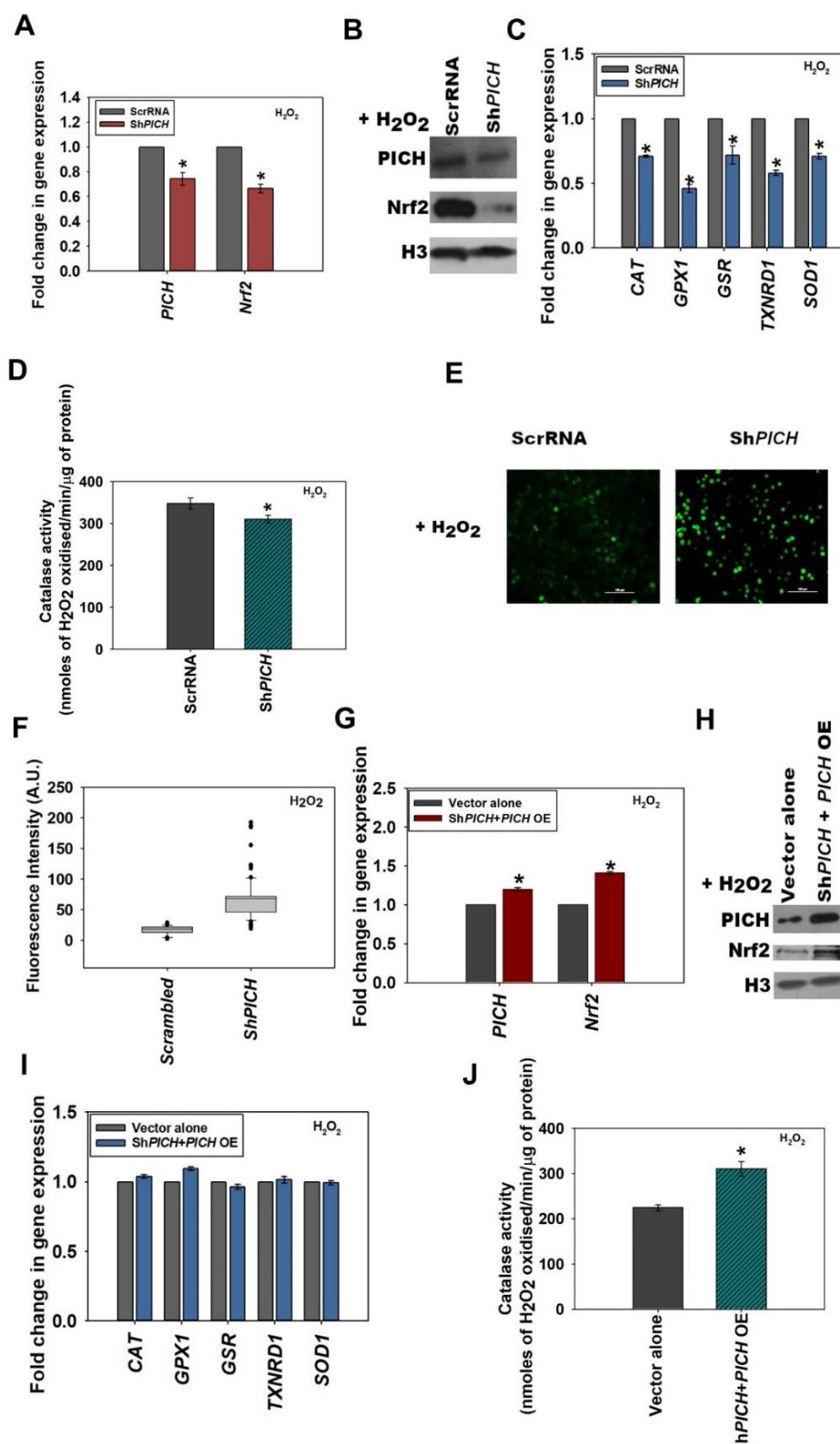

**Figure 2.** PICH regulates the expression of *Nrf2* in HeLa cells in the absence and presence of oxidative stress. (**A**) The expression of *PICH* and *Nrf2* in HeLa cells transfected with either ScrRNA or with Sh*PICH* plasmid in HeLa cells after treatment with 100 μM H$_2$O$_2$ for 20 min (*p*-value =2.21 × 10$^{-6}$ for Sh*PICH*). (**B**) The expression of PICH and Nrf2 in treated (100 μM H$_2$O$_2$; 20 min) HeLa cells transfected with either ScrRNA or with Sh*PICH*, analyzed by Western blot. (**C**) The expression of antioxidant genes *CAT*, *GPX1*, *GSR*, *TXNRD1*, and *SOD1* was quantitated with qRT-PCR in treated

(100 μM $H_2O_2$; 20 min) HeLa cells transfected either with ScrRNA or with Sh*PICH*. (**D**) Catalase activity (μmol/min) was quantitated in treated (100 μM $H_2O_2$; 20 min) HeLa cells transfected either with ScrRNA or with Sh*PICH*. (**E**) Cellular ROS was analyzed using DCFDA in treated HeLa cells transfected either with ScrRNA or with Sh*PICH* (*p*-value = 0.00). (**F**) Fluorescent intensity was quantitated using the software provided by TiE, Nikon Microscope. (**G**) Transcript levels of *PICH* and *Nrf2* were quantitated with qRT-PCR in treated (100 μM $H_2O_2$; 20 min) HeLa cells transfected with ScrRNA and empty vector or with the Sh*PICH* and *PICH* overexpression construct. (**H**) The expression of PICH and Nrf2 was analyzed with Western blot in treated (100 μM $H_2O_2$; 20 min) HeLa cells transfected with ScrRNA and empty vector or with the Sh*PICH* and *PICH* overexpression construct. (**I**) The expression of antioxidant genes *CAT*, *GPX1*, *GSR*, *TXNRD1*, and *SOD1* was quantitated in treated (100 μM $H_2O_2$; 20 min) HeLa cells transfected with ScrRNA and empty vector or with the Sh*PICH* and *PICH* overexpression construct. (**J**) Catalase activity (μmol/min) was estimated in treated (100 mM $H_2O_2$; 20 min) HeLa cells transfected with ScrRNA and empty vector or with the Sh*PICH* and *PICH* overexpression construct. *GAPDH* was used as the internal control in all the qRT-PCR experiments. The qRT-PCR experiments are presented as average ± SEM of three independent (biological) experiments with each biological experiment performed as two technical replicates. * Aasterisk indicates that *p*-value is < 0.05. The Western blots were repeated twice (biological independent experiments), and the best blot is represented in the figure. The catalase activity is presented as the average ± SEM of three independent experiments.

### 2.3. GAPDH Was Used as the Internal Control in the qRT-PCR Experiments

The qRT-PCR experiments are presented as average ± SEM of three independent (biological replicates with each replicate performed as two technical replicates) experiments. The Western blots are representative of three independent (biological replicates) experiments. The catalase activity is presented as the average ± SEM of three independent experiments.

The knockdown efficiency of *PICH* was 50–60% in all the experiments performed.

ROS levels, as analyzed by the DCFDA label, were found to be significantly upregulated in Sh*PICH* cells as compared with HeLa cells transfected with scrambled shRNA in both the absence and presence of $H_2O_2$ (Figures 2E,F and S2E,F).

*PICH* and Nrf2 expression was rescued when cells were transfected with Sh*PICH* along with a PICH overexpression cassette in both untreated and treated HeLa cells (Figures 2G–H and S2G–H; quantitation of Western blots provided in Figure S6G,H). The expression of antioxidant genes, as well as catalase activity, was also found to be rescued in $H_2O_2$-treated cells co-transfected with Sh*PICH* along with the *PICH* overexpression cassette, in contrast with cells transfected with vector alone (Figure 2I,J; untreated controls are shown in Figure S2I,J).

In all the subsequent experiments, we used Sh*PICH* shRNA for the downregulation of *PICH* expression.

### 2.4. Nrf2 Regulates the Expression of PICH in HeLa Cells in the Presence of Oxidative Stress

As the expression of PICH was upregulated on oxidative stress, we asked whether Nrf2 has a role in regulating the expression of this gene on oxidative stress. Nrf2 is a transcription factor that binds to antioxidant response element (ARE) and thus regulates the expression of antioxidant genes [28]. The polymorphic ARE core sequence 5′-RTGYCNNNGCR-3′, where R is a purine and Y is a pyrimidine, is present on the promoters of genes encoding for antioxidant genes [29,30]. An in silico analysis-identified ARE sequence was present on the *PICH* promoter, leading us to hypothesize that Nrf2 could potentially regulate the expression of this gene (Supplementary Table S3).

*Nrf2* expression was downregulated by transiently transfecting HeLa cells with shRNA against the 3′ UTR of the gene. Two shRNA were synthesized and labeled as Sh*Nrf2* and Sh*Nrf2_2*. The downregulation of *Nrf2* by Sh*Nrf2* led to the reduced expression of *PICH* in both the absence and presence of oxidative stress (Figure 3A–D; quantitation of western blots provided in Figure S6I,J). A similar result was obtained when cells were transfected with Sh*Nrf2_2* (Figure S3E–G; quantitation of western blot provided in Figure S6K). This

downregulation did not alter the expression of *DNA-PKc* in Sh*Nrf2* and Sh*Nrf2_2* cells in both the absence and in presence of oxidative stress (Figure S3H). The expression of PICH in *Nrf2*-downregulated cells could be rescued by overexpressing *Nrf2* in the presence of oxidative stress (Figure 3E–H; quantitation of Western blots provided in Figure S6L,M; heat maps shown in Figure S4).

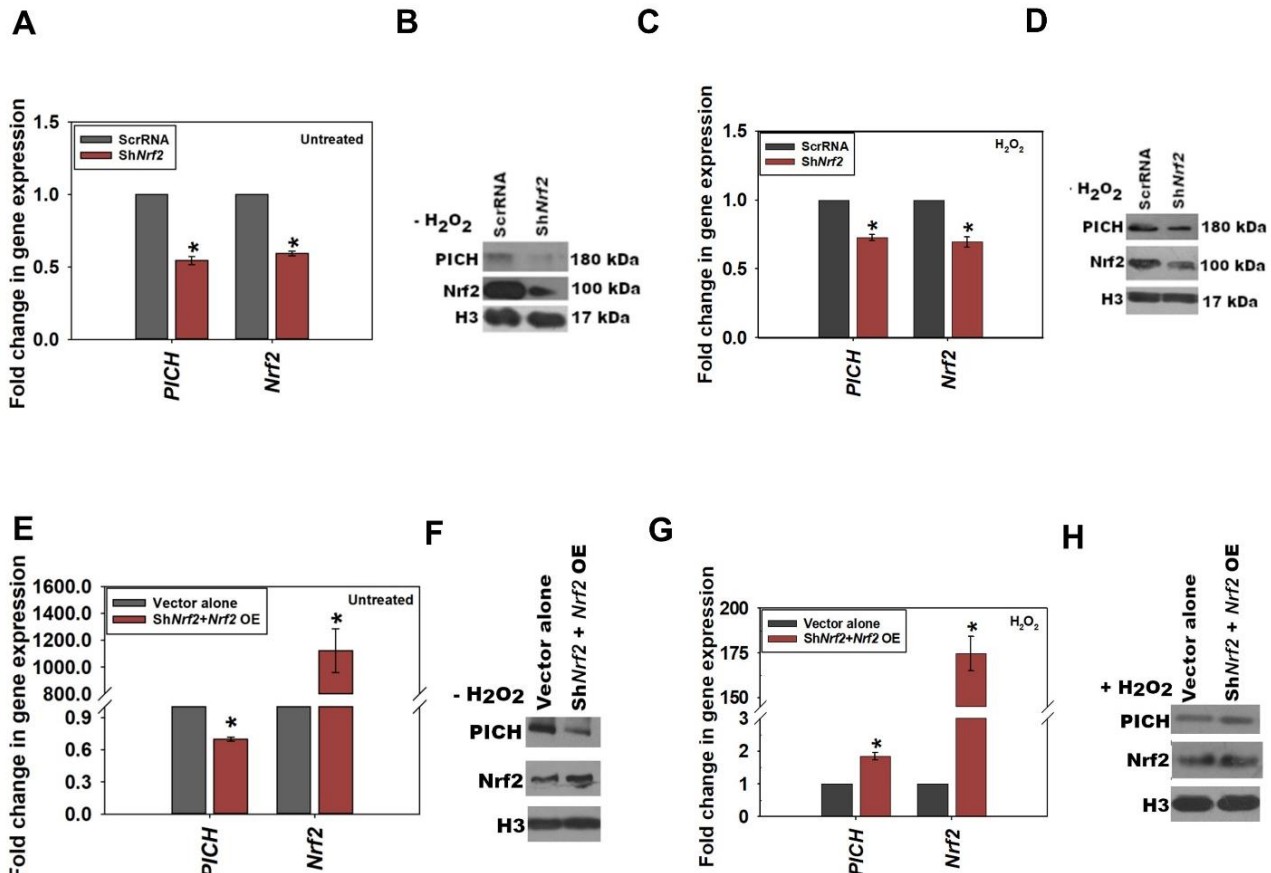

**Figure 3.** Nrf2 regulates the expression of *PICH* in HeLa cells in the presence of oxidative stress. (**A**) The expression of *PICH* and *Nrf2* in untreated HeLa cells transfected either with ScrRNA or with Sh*Nrf2* construct (*p*-value = 1.65 × 10⁻⁵). (**B**) The expression of PICH and Nrf2 was analyzed with Western blot in untreated HeLa cells transfected either with ScrRNA or with the Sh*Nrf2* construct. (**C**) The expression of *PICH* and *Nrf2* in treated (100 μM H₂O₂; 20 min) HeLa cells transfected either with ScrRNA or with Sh*Nrf2* construct (*p*-value = 3.56 × 10⁻⁶). (**D**) The expression of PICH and Nrf2 was analyzed with Western blot in treated (100 μM H₂O₂; 20 min) HeLa cells transfected either with ScrRNA or with Sh*Nrf2* construct. (**E**) Transcript levels of *PICH* and *Nrf2* were estimated by qRT-PCR in untreated HeLa cells transfected either with ScrRNA and empty vector or with Sh*Nrf2* construct along with *Nrf2* overexpression plasmid constructs. (**F**) The expression of PICH and Nrf2 were estimated by qRT-PCR in untreated HeLa cells transfected either with ScrRNA and empty vector or with Sh*Nrf2* construct along with *Nrf2* overexpression plasmid constructs. (**G**) Transcript levels of *PICH* and *Nrf2* were estimated using qRT-PCR in treated (100 μM H₂O₂; 20 min) HeLa cells transfected either with ScrRNA and empty vector or with Sh*Nrf2* construct along with *Nrf2* overexpression plasmid constructs. (**H**) The expression of PICH and Nrf2 were estimated using qRT-PCR in treated (100 mM H₂O₂; 20 min) HeLa cells transfected either with ScrRNA and empty vector or with Sh*Nrf2* construct along with *Nrf2* overexpression plasmid constructs. *GAPDH* was used as the internal control in the qRT-PCR experiments. The qRT-PCR experiments are presented as the average ± SEM of three independent experiments. The Western blots are representative of three independent experiments. The knockdown efficiency of *Nrf2* was 35–45% in all the experiments performed. * Asterisk indicates that *p*-value is < 0.05.

From these experimental results, it was concluded that in HeLa cells, Nrf2 expression correlates with the expression of *PICH* during oxidative stress.

Thus, *PICH* expression correlated with *Nrf2* expression in both the absence and presence of oxidative stress, while *Nrf2* expression appears to correlate with *PICH* expression during oxidative stress.

### 2.5. The Occupancy of PICH and Nrf2 Increases on the Promoters of Effector Genes on Oxidative Stress

To understand the mechanism of transcriptional regulation by PICH, ChIP experiments were performed to probe the occupancy of PICH, Nrf2, and RNAPII on *PICH, Nrf2, SOD1, GPX1*, and *TXNRD1* promoters in both the absence and presence of oxidative stress.

On the *PICH* promoter, PICH, Nrf2, and RNAPII occupancy increased only on oxidative stress (Figure 4A; heat map shown in Figure S5A). This suggests that Nrf2 does regulate the expression of *PICH* on oxidative stress. Further, PICH appears to co-regulate its own transcription.

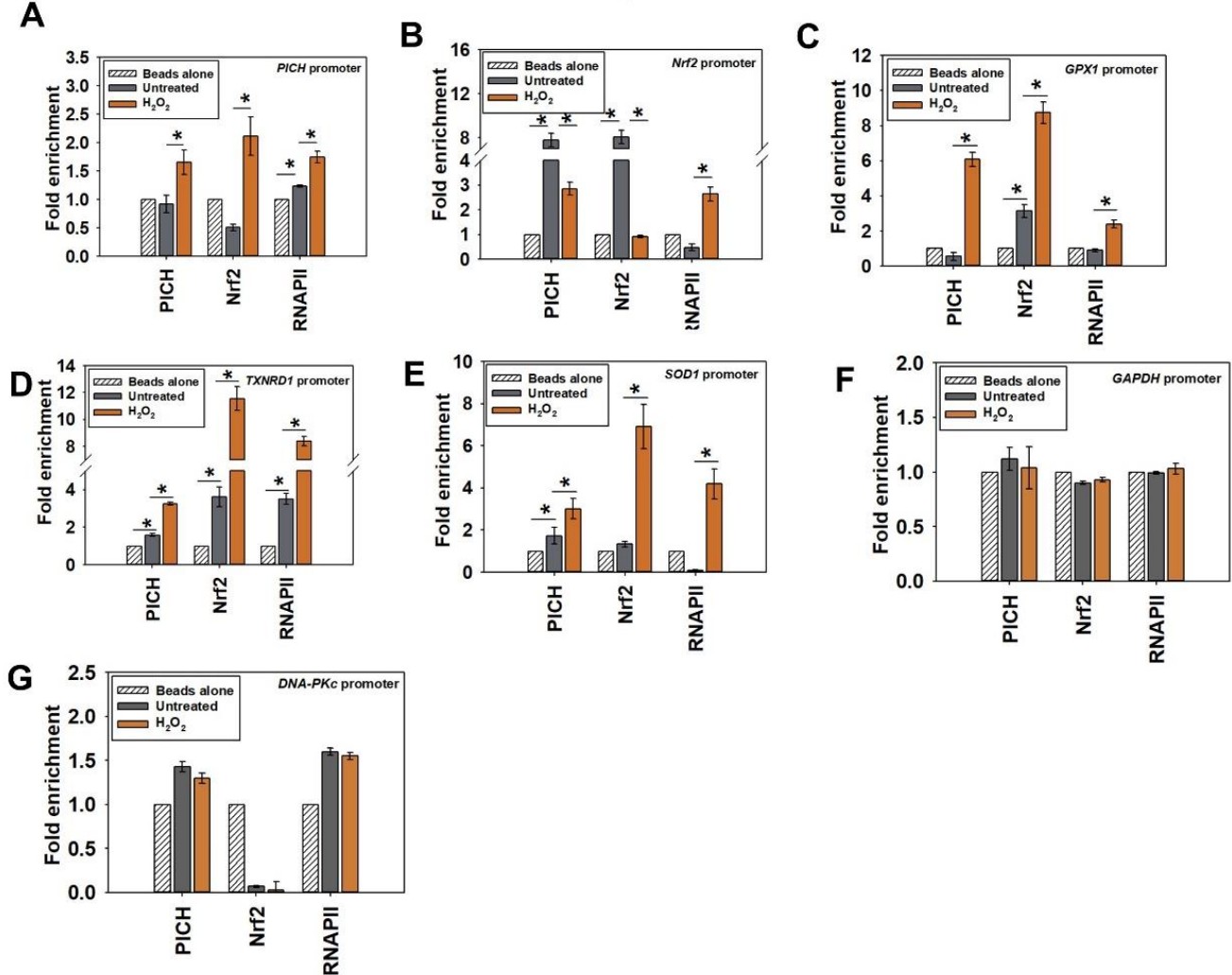

**Figure 4.** The occupancy of PICH and Nrf2 increases on the promoters of effector genes on oxidative stress. The occupancy of PICH, Nrf2, and RNAPII on the (**A**) *PICH* promoter; (**B**) *Nrf2* promoter; (**C**) *GPX1* promoter; (**D**) *TXNRD1* promoter; (**E**) *SOD1* promoter; (**F**) *GAPDH* promoter; and (**G**) *DNA-PKc* promoter were analyzed with ChIP in untreated and treated (100μM $H_2O_2$ for 20 min) HeLa cells. In these experiments, primers were made to probe a region −300 bp upstream and +200 bp downstream of the transcription start site (TSS). *GAPDH* promoter was used as the negative control. The ChIP data are presented as the average ± SEM of three independent experiments. * Asterisk indicates significance at *p*-value < 0.05.

PICH and Nrf2 were both found to be present on the *Nrf2* promoter; their occupancy decreased with oxidative stress as compared with the occupancy in the absence of oxidative stress (Figure 4B; heat map shown in Figure S5B). Overall, PICH regulates Nrf2 in both the absence and presence of oxidative stress. RNAPII occupancy, on the other hand, increased on the promoter of oxidative stress, correlating with the increased transcription of *Nrf2* (Figure 4B; heat map shown in Figure S5B).

Finally, PICH, Nrf2, and RNAPII occupancy increased on *GPX1, TXNRD1,* and *SOD1* promoters on oxidative stress, correlating with increased transcription (Figure 4C–E; heat map shown in Figure S5B).

In contrast, the occupancy of PICH, Nrf2, and RNAPII did not alter on *GAPDH* and *DNA-PKc* promoters (Figure 4F,G), indicating that the change in occupancy was specific to the genes involved in oxidative stress response.

Thus, PICH binds to the promoter (ARE) sequences of its target genes and co-regulates transcription with Nrf2.

Histone marks associated with transcription activation are enriched on PICH, Nrf2, and antioxidant gene promoters on oxidative stress: We hypothesized that the increased transcription of PICH, Nrf2, and antioxidant genes should be accompanied by an increase in activation histone marks. H3K9ac, H3K4me2, H3K4me3, and H3K27ac modifications are associated with transcription activation [31–33].

The analysis of the ChIP-seq data of these histone marks, H3K9c (GSM733756), H3K4me2 (GSM733734), H3K27ac (GSM733684), and H3K4me3 (GSM733682)—available in the public domain [31]—showed that these are present on the promoters of *PICH, Nrf2, GPX1, TXNRD1*, and *SOD1* in untreated HeLa cells (Figure S5A).

The CBP/p300 histone acetyltransferase has been shown to catalyze H3K27ac modification. Further, Nrf2 has been shown to co-immunoprecipitate with p300 [34]. Crosstalk between H3K27ac and H3K4me has also been reported [35]. Therefore, we decided to investigate the roles of these two modifications in the regulation of genes during oxidative stress. ChIP experiments were performed to understand the enrichment of H3K4me3 and H3K27ac on *PICH, Nrf2, GPX1, TXNRD1*, and *SOD1* promoters in the absence and presence of oxidative stress.

H3K4me3 and H3K27ac modifications were enriched on *PICH* promoter in the treated cells in contrast with untreated cells (Figure 5A). On the *Nrf2* promoter, however, the enrichment of only H3K27ac was observed in the treated cells in contrast with the untreated cells, indicating that only this modification is playing a role in the transcriptional regulation of this gene (Figure 5B). Finally, both H3K4me3 and H3K27ac modifications were found to be enriched on *GPX1, TXNRD1*, and *SOD1* promoters in the presence of oxidative stress, indicating that both these modifications are associated with the transcriptional activation of these genes (Figure 5C–E). In contrast, neither of these modifications was enriched on the *GAPDH* promoter (Figure 5F).

Thus, from these experiments, it was concluded that both H3K4me3 and H3K27ac were associated with the transcriptional regulation of *PICH* and antioxidant genes on oxidative stress. Further, only H3K27ac was playing a role in the transcriptional activation of the *Nrf2* gene on oxidative stress.

### 2.6. The Occupancy of Nrf2, RNAPII, and H3K27ac on the Target Genes Is Dependent on PICH Expression

Next, we investigated whether PICH is needed for the recruitment of Nrf2, RNAPII, and H3K27ac to the promoters of the target genes on oxidative stress. ChIP experiments were performed using cells transfected with either scrambled shRNA (ScrRNA) or shRNA against the 3′ UTR of *PICH* (Sh*PICH*) and treated with $H_2O_2$.

The occupancy of PICH, RNAPII, and H3K27ac decreased on *PICH, Nrf2, GPX1, TXNRD1*, and *SOD1* promoters in the Sh*PICH* cells, in contrast with, ScrRNA cells on oxidative stress correlating with decreased expression of these genes (Figure 6A–E). Nrf2 occupancy also decreased on *PICH, SOD1, GPX1*, and *TXNRD1* promoters; however, it increased on its own promoter in Sh*PICH* cells, in contrast with ScrRNA on oxidative stress

(Figure 6A–E). In contrast, the occupancy of PICH, Nrf2, RNAPII, and H3K27ac did not alter either *GAPDH* or *DNA-PKc* promoters (Figure 6F,G).

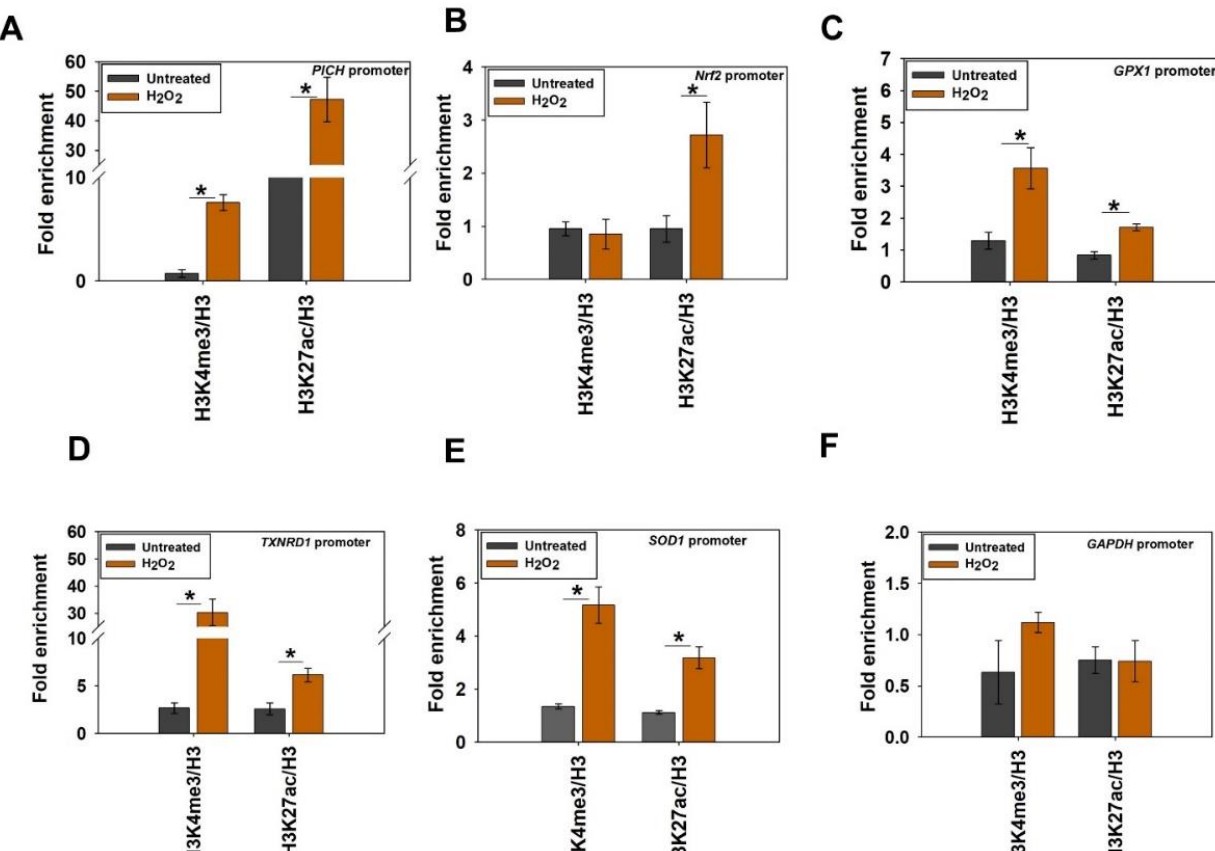

**Figure 5.** Histone marks associated with transcription activation are enriched on PICH, Nrf2, and antioxidant gene promoters on oxidative stress. The fold enrichment of H3K4me3 and H3K27ac as a ratio of H3 was determined using ChIP on the (**A**) *PICH* promoter; (**B**) *Nrf2* promoter; (**C**) *GPX1* promoter; (**D**) *TXNRD1* promoter; (**E**) *SOD1* promoter; and (**F**) *GAPDH* promoter in untreated and treated (100 μM $H_2O_2$, 20 min) HeLa cells. In these experiments, primers were made to probe a region −300 bp upstream and +200 bp downstream of the transcription start site (TSS). *GAPDH* promoter was used as the negative control. The ChIP data are presented as average ± SEM of three independent (biological replicates) experiments. * Aasterisk indicates significance at *p*-value < 0.05.

Taken together, we hypothesize that PICH is needed for the recruitment of transcription machinery to the target genes on oxidative stress.

### 2.7. PICH and Nrf2 Are Present Simultaneously on the Antioxidant Gene Promoters

Next, we asked whether PICH is present on the target promoters simultaneously with Nrf2. ChIP-reChIP experiments confirmed that PICH and Nrf2 are indeed present together in the promoter regions of *GPX1*, *TXNRD1*, and *SOD1* (Figure 6H–J).

Therefore, we concluded that PICH and Nrf2 are simultaneously present on the promoters of the target genes.

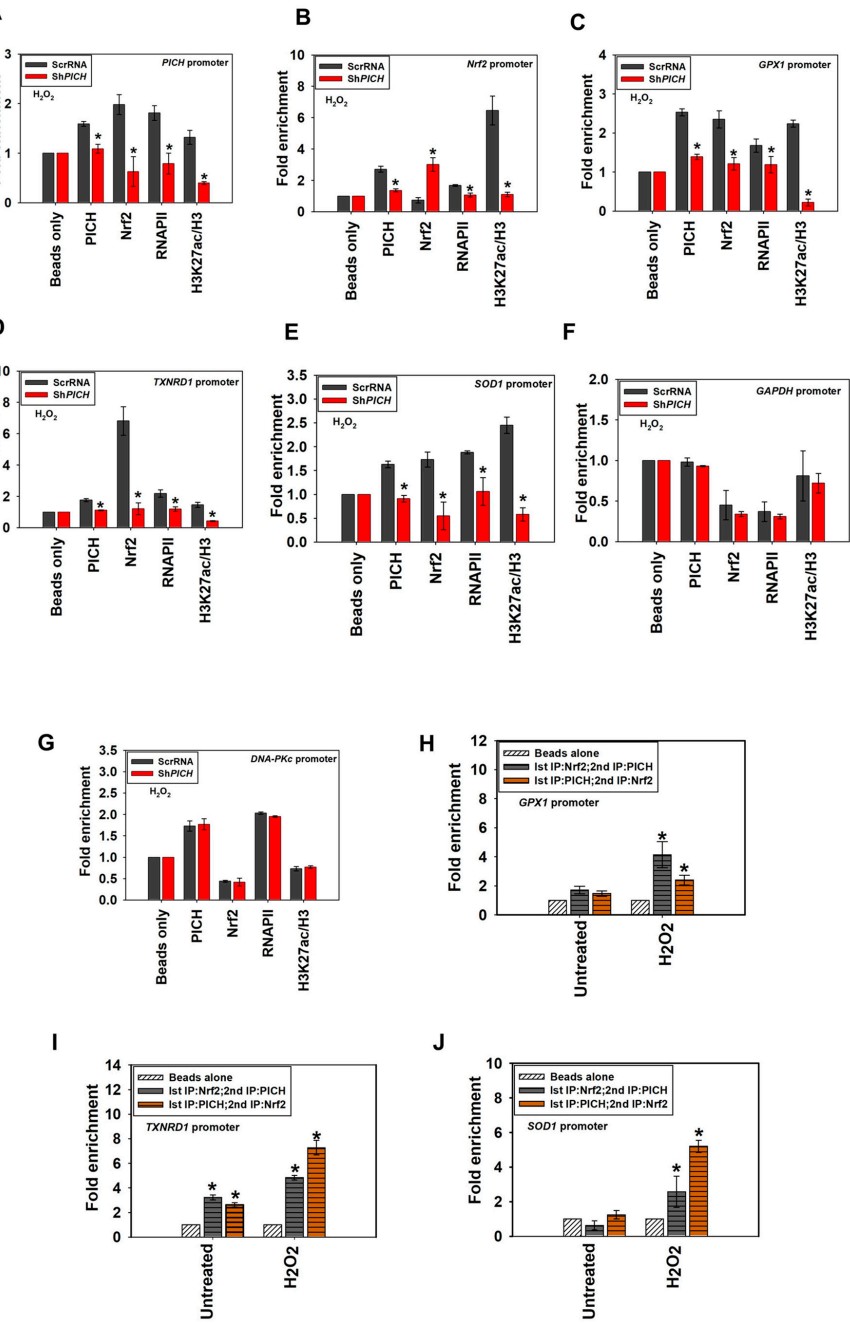

**Figure 6.** The occupancy of Nrf2, RNAPII, and H3K27ac on the target genes is dependent on *PICH* expression. The occupancy of PICH, Nrf2, and RNAPII, as well as the fold enrichment of H3K27ac as a ratio of H3, was assessed using ChIP in HeLa cells transfected either with ScrRNA or Sh*PICH* after treatment with 100 μM $H_2O_2$ for 20 min on (**A**) *PICH* promoter; (**B**) *Nrf2* promoter; (**C**) *GPX1* promoter; (**D**) *TXNRD1* promoter; (**E**) *SOD1* promoter; (**F**) *GAPDH* promoter; and (**G**) *DNA-PKc* promoter. The simultaneous occupancy of PICH and Nrf2 was assessed in untreated and treated HeLa cells using ChIP re-ChIP on (**H**) *GPX1* promoter; (**I**) *TXNRD1* promoter; and (**J**) *SOD1* promoter. In these experiments, primers were made to probe a region −300 bp upstream and +200 bp downstream of the transcription start site (TSS). The ChIP data are presented as the average ± SEM of three independent experiments. A star indicates significance at $p$-value < 0.05. In the ChIP-reChIP experiments, primers were made to probe a region −300 bp upstream and +200 bp downstream of the transcription start site (TSS). The ChIP-reChIP data for *SOD1*, *GPX1*, and *TXNRD1* promoters are presented as average ± SEM of two independent experiments. * Asterisk indicates significance at $p$-value < 0.05.

### 3. Discussion

Oxidative stress reflects the imbalance between the expression of reactive oxygen species and the ability of the cell to detoxify or repair the damage through reactive intermediates. The reactive oxygen species cause DNA damage, lipid peroxidation, and the oxidation of proteins. The cell combats the oxidative stress by producing antioxidants like superoxide dismutase, catalase, and glutathione peroxidase that can directly scavenge the free radicals. These antioxidant enzymes are expressed under transcriptional control, wherein Nrf2, a transcription factor, has been shown to play a pivotal role. In addition, studies have shown that the expression of antioxidant genes is also co-regulated by chromatin remodeling mechanisms. KMT2D, a histone methyltransferase that catalyzes the monomethylation of H3K4, has been shown to regulate the expression of antioxidant genes in prostate cancer [36]. Studies have also shown that the transcriptional activation of *HO-1*, a gene encoding for homo oxygenase -1, by Nrf2 requires BRG1, an ATP-dependent chromatin remodeling protein [3]. However, there are no reports regarding the role of PICH in mediating the transcriptional co-regulation of genes encoding for antioxidants during oxidative stress.

PICH was identified as a PLK-1 kinase-interacting protein, and extensive studies have delineated its role in mitosis [5,37,38]. Further, studies have shown that PICH is an ATP-dependent translocase that can bind and translocate on double-strand DNA but cannot remodel nucleosomes [8]. Thus, the role of this protein in transcriptional co-regulation has not yet been investigated.

In this paper, the role of PICH as a transcriptional co-regulator when oxidative stress is generated in HeLa cells has been delineated. PICH expression correlated with the expression of *Nrf2* as well as genes encoding for antioxidants in HeLa cells on oxidative stress. The downregulation of *PICH* reduced the expression of *Nrf2* as well as genes encoding for antioxidants, leading to increased oxidative stress on treatment with $H_2O_2$. The overexpression of wild-type *PICH* in *PICH*-depleted cells restored the expression of *Nrf2* and antioxidants in the presence of oxidative stress. ChIP analysis confirmed that PICH can indeed bind to the promoter regions of Nrf2 and antioxidant genes, thus possibly directly regulating their expression.

The expression of *PICH* appears to be regulated by Nrf2 in the presence of oxidative stress. In silico analysis identified ARE sequence in the promoter regions of *PICH*, and ChIP experiments confirmed that Nrf2 occupancy increases on the *PICH* promoter only in the presence of oxidative stress. In contrast, PICH appears to regulate the expression of *Nrf2* in both the absence and presence of oxidative stress. Thus, PICH provides cellular defense by functioning as an oxidative stress-induced response factor that regulates the transcription of Nrf2 and antioxidant genes.

The analysis of the available ChIP-seq datasets showed that activating histone marks H3K9ac, H3K4me2, H3K27ac, and H3K4me3 are enriched on *PICH*, *Nrf2*, *GPX1*, *TXNRD1*, and *SOD1* promoters. ChIP experiments confirmed that H3K27ac and H3K4me3 are enriched on the *PICH*, *GPX1*, *TXNRD1*, and *SOD1* promoters on oxidative stress, while only H3K27ac is enriched on the *Nrf2* promoter. Thus, on oxidative stress, gene expression is regulated by Nrf2, a transcription factor, possibly in collaboration with the epigenetic regulators PICH and activating histone modifications, leading to an open chromatin architecture and thus, increased transcription.

The depletion of *PICH* led to the reduced enrichment of H3K27ac, RNAPII, and Nrf2 on the promoters of the responsive genes, suggesting that PICH is needed for the recruitment of the transcription machinery on oxidative stress. However, the order of recruitment cannot be delineated from these experiments as the expression of Nrf2 is affected in PICH-depleted cells. Based on the results, we propose three models to explain the role of PICH in regulating transcription (Figure 7). In the first model, PICH is recruited first to the promoter regions. Subsequently, it recruits Nrf2 and the histone-modifying enzymes. This results in the opening of the chromatin. Alternatively, Nrf2 and histone modifications occur first. Nrf2/histone modifications help in recruiting PICH. This results

in the opening of the chromatin. The third possibility involves PICH-Nrf2 along with the histone-modifying enzyme being present as a complex. The complex is recruited to the promoter, where it mediates chromatin remodeling. The studies presented here do not allow us to differentiate between these different mechanisms because PICH depletion leads to the downregulation of Nrf2. Additional experiments need to be performed to understand how these factors are recruited to the target genes on oxidative stress.

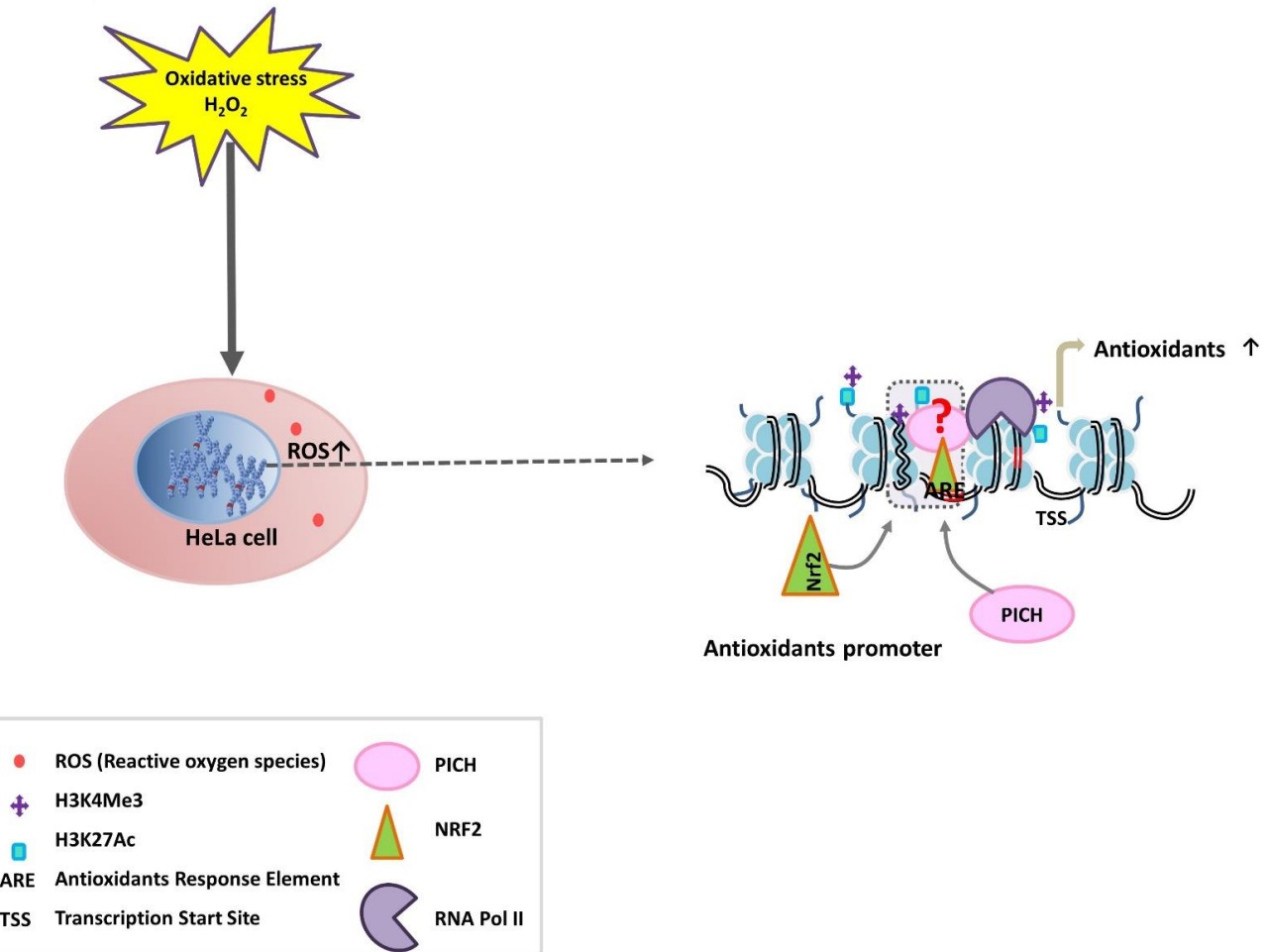

**Figure 7.** Models to explain the role of PICH in modulating the expression of antioxidant genes. Following $H_2O_2$ (oxidative stress) treatment, the amount of ROS increases inside the cells. PICH, Nrf2, and histone modifications regulate the expression of antioxidant genes. Three possibilities exist: (i) PICH is recruited to the promoter first followed by Nrf2 and modification of H3; (ii) Nrf2 is recruited first leading to the modification of histones and recruitment of PICH; (iii) Nrf2, PICH, and histone-modifying enzymes are present as a complex and are recruited together to the promoter region.

Oxidative stress is lethal to cells, and thus, cells have devised mechanisms to alleviate it. Regardless of the model, PICH appears to be central to this process by regulating the expression of both antioxidant genes and Nrf2, the transcription factor involved in regulating the expression of antioxidant genes, thus expanding the role of this protein beyond mitosis into the realm of transcription regulation.

## 4. Materials and Methods

### 4.1. Chemicals

DMEM, fetal bovine serum, antibiotic antimycotic solution, and trypsin-EDTA solution were purchased from Himedia (Thane West, Maharashtra, India). Hoechst 33342 and Trizol

were purchased from Sigma-Aldrich (Burlington, MA, USA). Hydrogen peroxide was purchased from Merck (Darmstadt, Germany). RevertAid First Strand cDNA Synthesis Kit and TurboFect transfection Reagents were purchased from Thermo Fisher Scientific (Waltham, MA, USA). Restriction endonucleases were purchased from New England Biolabs (Ipswich, MA, USA). SYBR Green PCR Master Mix was purchased from Kapa Biosystems (Wilmington, MA, USA). Micro-amp Fast 96-well reaction 0.1 mL plates were purchased from Applied Biosystems (Waltham, MA, USA). QIAquick gel extraction kit was purchased from Qiagen (Hilden, Germany). Immobilon-P PVDF membranes were purchased from Merck-Millipore (Burlington, MA, USA). X-ray films, developer, and fixer were from Kodak (Rochester, NJ, USA). DAAO plasmid [26] was a kind gift from S.K. Goswami, School of Life Sciences, Jawaharlal Nehru University (New Delhi, India).

### 4.2. Primers

Primers for ChIP and qRT-PCR as well as oligonucleotides used for biophysical studies were synthesized either by Sigma-Aldrich (Burlington, MA, USA) or by GCC Biotech (Joychandipur, West Bengal, India). The sequence of the primers and oligonucleotides used in this study is provided in Supplementary Tables S1–S4.

### 4.3. Antibodies

BRG1 (catalog#Ab70558), histone H3 (catalog# Ab1791), RNAPII (Rpb1 CTD catalog# 2629), Nrf2 (catalog#Ab62352), PICH (catalog# Ab88560), H3K27ac (catalog#4729) antibodies were purchased from Abcam (Cambridge, UK). For Western blot, Nrf2 (catalog# 16396-1-AP) was purchased from Proteintech Group (Rosemont, IL, USA). H3K4me3 (catalog# C42D8) and IgG (catalog# 2729) were purchased from Cell Signaling Technology (Danvers, MA, USA).

### 4.4. Transient Downregulation of PICH, Nrf2

*PICH* and *Nrf*2 were downregulated in HeLa cells using shRNA constructs against the 3′ UTR (Supplementary Table S3). Two short regions of 21 bases from the 3′ UTR were selected as the sense fragments for *PICH* shRNA. Similarly, two short regions from the 3′ UTR were selected as the sense fragments for *Nrf*2. These were ligated to the antisense fragment interrupted by a non-related spacer sequence. The double-stranded oligonucleotide encoding the shRNA was cloned into the pLK0.1 vector between AgeI and EcoRI restriction sites.

### 4.5. Overexpression of PICH and Nrf2

The enhanced GFP tagged construct for *PICH*, pEGFP PICH (Nigg CB62) was a gift from Erich Nigg (Addgene plasmid # 41163; http://n2t.net/addgene:41163 (accessed on 24 May 2021); RRID: Addgene_41163) [5].

pcDNA3-EGFP-C4-Nrf2 was a gift from Yue Xiong (Addgene plasmid # 21549; http://n2t.net/addgene:21549 (accessed on 24 May 2021; RRID: Addgene_21549) [39]

### 4.6. Cell culture and Transfection

HeLa cells obtained from NCCS, Pune, India were cultured in DMEM containing 10% fetal bovine serum and 1× antibiotic antimycotic solution at 37 °C and 5% $CO_2$. For inducing oxidative stress, the cells were treated with 100 μM hydrogen peroxide for a specified time.

### 4.7. RNA Isolation and qRT-PCR

Total RNA was extracted using the Trizol reagent. 90% confluent cells in a 35 mm plate were lysed with 1 mL of the Trizol reagent to give a homogenized lysate. The lysate was transferred to an Eppendorf tube. 200 μL of chloroform was added to each tube per ml of Trizol reagent, shaken vigorously, and allowed to stand for 10–15 min at room temperature. The samples were centrifuged at 11,000 rpm for 15 min at 4 °C. The top aqueous layer

was transferred to a fresh Eppendorf tube, and 0.5 mL of isopropanol was added per ml of Trizol reagent, mixed, and allowed to stand at room temperature for 10–15 min. The samples were then centrifuged at 11,000 rpm for 10 min at 4 °C. The RNA pellet obtained was washed with 70% ethanol and resuspended in DEPC-treated water. The concentration of the purified RNA was determined using NanoDrop 2000 (Thermo Fisher Scientific, USA) and an equal amount of RNA from various samples was used for preparing the cDNA using random hexamer primers according to the manufacturer's protocol. The prepared cDNA was checked for quality by performing a PCR using suitable primers.

Quantitative real-time RT-PCR (qRT-PCR) was performed using the 7500 Fast Real-Time PCR system (ABI Biosystems, Waltham, MA, USA). Gene-specific primers designed for exon-exon junctions were used for qRT-PCR. For each reaction, 10 μL of samples were prepared in triplicate, and the data obtained were analyzed using Fast7500 software V 2.06 provided by the manufacturer.

### 4.8. Chromatin Immunoprecipitation (ChIP)

The cells were cross-linked for 10 min by adding formaldehyde (final concentration 1%) and later quenched by adding glycine (final concentration 125 mM) to the medium. The cells were then washed thoroughly using ice-cold PBS and scraped into 1 mL buffer containing 150 mM NaCl, 0.02 M EDTA, 50 mM Tris-Cl (pH 7.5), 0.5% (*v/v*) NP-40, 1% (*v/v*), Triton-X100, and 20 mM NaF. The cells were collected in Eppendorf tubes and pelleted at 12,000 g at 4 °C for 2 min twice. The pelleted cells were treated with freshly prepared 150 μL of sonication buffer 1 containing 0.01 M EDTA pH 8.0, 50 mM Tris-Cl (pH 8.0), and 1% SDS for 15 min at 4 °C on a rocker. This was followed by the addition of 50 μL of freshly prepared sonication buffer 2 containing 30 M EDTA 40 mM Tris-Cl (pH 8.0) 2% (*v/v*) NP-40, 0.04% (*w/v*) NaF. Sonication was performed using a water bath sonicator (40 cycles of 30 s pulse/20 s rest). The sonicated samples were centrifuged, and the supernatant was used for further analysis. 50 μL of the sonicated sample was purified, and DNA concentration was determined. An equal amount of chromatin (25 μg) was taken for performing IP using various antibodies. One sample was kept as beads alone for negative control. Pre-adsorbed protein A bead resin (pre-adsorbed with 75 ng/μL sonicated salmon sperm DNA and 0.1 μg/μL of BSA) and 1.5 to 2 μg of the desired antibody were added to each sample and incubated overnight at 4 °C on a rotator. This was followed by washing the pelleted bead resin two times in IP buffer containing 0.15 M NaCl, 0.02 M EDTA (pH 8.0), 50 mM Tris-Cl (pH 8.0), 1% (*v/v*) NP-40, 0.02% NaF, 0.50% sodium deoxycholate and 0.1% SDS. This was followed by washing three times in wash buffer (0.5 M LiCl, 0.02 M EDTA (pH 8.0), 0.1 mM Tris-Cl (pH 8.0), 1% (*v/v*) NP-40, 0.02% NaF and 1% sodium deoxycholate). The immune complexes were again washed twice in IP buffer, and then a final wash was given in TE buffer (0.01 M Tris-Cl (pH 8.0), 0.001 M EDTA). The bound DNA was eluted using 100 μL of 10% (*w/v*) Chelex® 100 (Bio-Rad, USA) slurry prepared per the manufacturer's instructions. The eluted DNA was used for qRT-PCR using standardized primers.

### 4.9. Western Blotting

Adherent cells in 100 mm dishes were washed with 1X PBS 3–4 times, scraped with a cell scraper, and collected in 1 mL of 1X PBS. The cells were pelleted at 2500 rpm at 4°C. The cell lysate was prepared on ice using 300 μL of modified RIPA lysis buffer (50 mM Tris-Cl pH7.5, 150 mM NaCl, 2 mM EDTA, 0.5% sodium deoxycholate, 1% (*v/v*) NP-40, 0.1% sodium dodecyl sulphate, 1% TritonX-100, and 3 mM PMSF). The samples were sonicated for 3 cycles (20 s ON,40 s OFF) using a sonicator. The protein concentration was determined using the Bradford assay. 100 μg protein was used to run the SDS-PAGE. Western blotting was performed as described in (https://www.abcam.com/protocols/general-western-blot-protocol#loading-and-running-the-gel (accessed on 22 June 2021)). The western blots were quantitated using Image J software (Version ImageJ1.8.0).

### 4.10. Oxidative Stress Assay

DCFDA/H2DCFDA/DCFH-DA/DCFH is a fluorogenic dye that detects hydroxyl, peroxyl, and other reactive oxygen species (ROS) activity within a cell. The DCFDA assay protocol is based on the diffusion of DCFDA/H2DCFDA/DCFH-DA/DCFH into the cell. Inside the cells, this molecule is deacetylated by cellular esterases to a non-fluorescent compound. This non-fluorescent molecule is oxidized by ROS into $2',7'$–dichlorofluorescein (DCF), a highly fluorescent compound that can be detected with fluorescence spectroscopy at excitation/emission of 485 nm/535 nm.

DCFDA (Thermo Fisher Scientific, Waltham, MA, USA) was prepared as a 10 mM stock in DMSO. Cells were seeded in a 35 mm dish to reach 40% to 50% confluency on the day of the experiment. The serum containing DMEM medium was replaced with serum-free DMEM (SFM) medium for 6 h before the start of the experiment to reduce the level of serum available in the medium.

At the time of the experiment, 10 μM DCFDA was added to the cells in dark and incubated at 37 °C for 10 min. Subsequently, SFM containing 100 μM $H_2O_2$ was added, and the cells were further incubated for 20 min. Fluorescence was recorded for at least 100 cells using a fluorescent microscope (Eclipse Ti-E, Nikon, Tokyo, Japan) at the FITC range of 495 nm.

### 4.11. Chip-reChip

ChIP-reChIP was performed as described in [40].

### 4.12. Catalase Activity Assay

The cell lysate was prepared on ice using 300 μL of modified RIPA lysis buffer (50 mM Tris-Cl (pH 7.5), 150 mM NaCl, 2 mM EDTA, 0.5% sodium deoxycholate, 1% (*v/v*) NP-40, 0.5% sodium dodecyl sulphate, 1% TritonX-100, and 3 mM PMSF). Subsequently, 0.95 mL of 0.1 M phosphate buffer (pH 7.4), 1.0 mL of freshly prepared 0.05 M hydrogen peroxide, and 100 μg supernatant were added in a 3 mL cuvette. The optical density was read at 240 nm using Cary 60 UV-Vis spectrophotometer (Agilent Technologies, Santa Clara, CA, USA). The activity was calculated as follows:

$$Catalase\ Activity\ (\text{nmoles of } H_2O_2 \text{ consumed/min/mg protein}) =$$
$$\Delta OD/\min \times Volume\ of\ Assay \times 10^9/MEC \times Volume\ of\ Enzyme \times \text{mg protein} \times PL \times VCF$$

where: *Volume of Assay* = 3.0 mL, *MEC* = Molar Extinction Coefficient = 39.6 $M^{-1} \cdot cm^{-1}$, *Volume of Enzyme* = 0.05 mL, *VCE* = Volume Conversion Factor = 100, *PL* = Path Length = 1 cm.

### 4.13. Statistical Analysis

All qRT-PCR and ChIP experiments are reported as the average ± standard error of the mean (SEM) of three independent (biological) experiments unless otherwise specified. Each independent experiment was performed as at least two technical replicates. As we have analyzed data from biological replicates, we have calculated SEM instead of standard deviation (SD). The statistical significance (*p*-value) was calculated using the paired t-test available in Sigma Plot. The differences were considered significant at $p < 0.05$.

### 4.14. Quantitation of Western Blots

All the quantitation data of the Western blots are provided in Supplementary Figure S6.

### 4.15. Original Blots

The original blots are provided in Supplementary Figure S7.

**Supplementary Materials:** The following supporting information can be downloaded at: https://www.mdpi.com/article/10.3390/epigenomes6040036/s1, Figure S1: *PICH* expression is upregulated when cells are exposed to oxidative stress, Figure S2: PICH regulates the expression of *Nrf2* in HeLa cells in the absence and presence of oxidative stress, Figure S3: *ShPICH_2* regulates the expression of *Nrf2* in HeLa cells in the absence and presence of oxidative stress, Figure S4: Heat maps of the fold changes found in qRT-PCR analysis, Figure S5: Histone marks associated with transcription activation are enriched on PICH, Nrf2, and antioxidant gene promoters on oxidative stress, Figure S6: Quantification of all western blots, Figure S7: Original western blots, Table S1: List of primers used in qRT-PCR experiments, Table S2: List of primers used in ChIP experiments, Table S3: List of target sequences used in making ShRNA constructs, Table S4. Primer used for making PICH K128A mutant.

**Author Contributions:** Conceptualization, R.M.; Formal analysis, A.D. (Anindita Dutta); Funding acquisition, R.M.; Investigation, A.D. (Anindita Dutta), A.D. (Apurba Das), D.B. and V.A.; Methodology, A.D. (Anindita Dutta), A.D. (Apurba Das) and R.M.; Supervision, R.M.; Writing—original draft, A.D. (Anindita Dutta) and R.M. All authors have read and agreed to the published version of the manuscript.

**Funding:** R.M. was supported by grants from SERB (EMR/2015/002413 and CRG/2020/000607), India. A.D. was supported by a fellowship from CSIR. D.B. was supported by a UGC non-NET fellowship. The authors would also like to acknowledge funding from DST-FIST.

**Data Availability Statement:** The data that support the findings of this study are available from the corresponding author upon reasonable request.

**Acknowledgments:** The authors would like to thank the Central Instrumentation Facility, School of Life Sciences for the fluorescence microscope.

**Conflicts of Interest:** The authors declare they have no conflicts of interest.

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
