# Peer review of "PLK-1 Interacting Checkpoint Helicase, PICH, Mediates Cellular Oxidative Stress Response"

_2075-4655_

Round 1
Reviewer 1 Report
Overall, this is a clear, concise, and well-written manuscript. The methods used and the rationale is appropriate for demonstrating the role of PICH protein. The introduction is a little bit shorter and can be extended to give a background on the importance of the study that will be easier to understand for the readers about why the study is necessary.
Author Response
Reviewer#1
Comments and Suggestions for Authors
Overall, this is a clear, concise, and well-written manuscript. The methods used and the rationale is appropriate for demonstrating the role of PICH protein. The introduction is a little bit shorter and can be extended to give a background on the importance of the study that will be easier to understand for the readers about why the study is necessary.
Our Response: We thank the reviewer for appreciating our work. We have rewritten the introduction part in the revised manuscript and mentioned why this study is necessary.
Reviewer 2 Report
The manuscript “PICH, an ATP-dependent Chromatin remodelling protein transcriptionally co-regulates oxidative stress response” by Dutta et al attempts to bring light into the molecular players involved in the oxidative stress response. The manuscript contains a lot of experiments, really lots, that unfortunately do very little to support their overall claims. The abstract is not clear and the introduction rather enumerates things without a clear directionality. The overall quality of the manuscript is rather low as the main text (all of it), figures (low quality), logics and presentation of the results fails to convey an understandable message.
Major comments:
1- The abstract and the intro do very little to introduce both state-of-the-art, context and direction of the study. I suggest the authors to thoroughly revise both sections for future iterations of the manuscript.
2- There are overstatements all over the manuscript. I suggest the authors to tone down their findings as by no means are formally demonstrated here.
3- The authors assess all over the manuscript transcript levels by using reverse transcription followed by qPCR. However, they state that they use qPCR. Not the same thing.
4- The authors do not describe the rationale behind many of the experiments and just throws data out of the blue. Please have a look and give context and the explain the reasons underlying each experimental decision. For instance: why measuring gammaH2AX? Why is it confirmatory?
5- The authors suggest in multiple places of the manuscript that they confirm their (RT-) qPCRs by Western blot. This is an absolutely wrong statement as mRNAs are not translated just because there is more of them. Multiple mechanisms operate over mRNA translation and stability including mRNAs that actually do different things than just being mere messengers; thus, the claim is not valid and must not be used.
6- In multiple places the authors provide statistical test’s information within the main text. This must be removed and placed in figure legends for each figure.
7- Multiple data comparisons are scattered all over the figures and supplementary figures. I suggest the authors to establish valid comparisons within the same figures and to collect comparable data in single graphs rather than spanning them all across the manuscript.
8- Many experiments add very little to the (I believe) overall message. For instance, the microscopy assay or the Western blot quantifications. All this should be either removed or send to supplementary.
9- I suggest the authors to include the values of untreated/untransfected for every time point in a time course. Otherwise, the displayed data is not convincing.
10- All the fragment regarding cell line specificity is rather mute. Should be removed as it does not bring anything to the table. Especially when only 2 cell lines are compared.
11- Please make the figures auto-explicative by for instance replacing the word “treated” with H2O2.
12- How does the authors normalize their assays for transfection efficiency? Without this crucial piece of information, no conclusion can be made out of any of the transfection experiments.
13- P-Values mean very little if no context is given for the statistical test. Is it an ANOVA, a two-tailed t-test, what is it? All this info must be included in the figure legend.
14- Antibody validation by intracellular localization as determined by microscopy is a no go.
15- The authors suggest that PICH regulates Nrf2 but the data displayed only supports a correlation. Please tone this down.
16- The wording is too strong in some places. For instance: Line 459… Nrf2 regulates the expression of PICH “only” on oxidative stress.
17- Authors should show IgG instead of beads in their ChIP data for all ChIP data as the control values are not reported in half of the document.
18- Why the authors only measure 2 histone modifications??? They suggest that there may be 4 different histone modifications or even more by the restrict their analysis to only 2. Moreover, H3K27Ac is often found in enhancers not promoters albeit a few reports. This clearly needs to be discussed.
19- The biochemical/structural analysis is completely out of place in here and the data is not sufficient to draw any conclusion. I suggest the authors to focus and what is to be the main message of the manuscript and display the data accordingly rather than tossing all together and trying to make it work.
20- The discussion is way too long for the contents of the manuscript. The major and minor conclusions of the manuscript are not substantiated by the data and it is the opinion of this reviewer that the manuscript needs a major overhaul for it to work minimally as fundamental controls and experimental evidence are missing.
Minor comments
1- Please check manuscript wording and syntax for a future iteration.
2- All figures are displayed with low resolution. I suggest the authors to fix this.
3- Place comparisons in single charts to enable direct comparison. I have the impression that all the data may be condensed in 2-3 figures.
4- Please work on the logics all over the manuscript as the way the data is presented is not convincing at all.
Author Response
Reviewer#2
Comments and Suggestions for Authors
The manuscript “PICH, an ATP-dependent Chromatin remodelling protein transcriptionally co-regulates oxidative stress response” by Dutta et al attempts to bring light into the molecular players involved in the oxidative stress response. The manuscript contains a lot of experiments, really lots, that unfortunately do very little to support their overall claims. The abstract is not clear and the introduction rather enumerates things without a clear directionality. The overall quality of the manuscript is rather low as the main text (all of it), figures (low quality), logics and presentation of the results fails to convey an understandable message.
Major comments:
- The abstract and the intro do very little to introduce both state-of-the-art, context and direction of the study. I suggest the authors to thoroughly revise both sections for future iterations of the manuscript.
Our Response: We have rewritten the abstract and the introduction part in the revised manuscript.
- There are overstatements all over the manuscript. I suggest the authors to tone down their findings as by no means are formally demonstrated here.
Our Response: We have attempted to tone down the significance of our findings as suggested by the reviewers in the revised manuscript. If it needs to be further toned down, we can do so.
- The authors assess all over the manuscript transcript levels by using reverse transcription followed by qPCR. However, they state that they use qPCR. Not the same thing.
Our Response: As suggested by the reviewer, we have changed qPCR with qRT-PCR in the revised manuscript.
- The authors do not describe the rationale behind many of the experiments and just throws data out of the blue. Please have a look and give context and the explain the reasons underlying each experimental decision. For instance: why measuring gammaH2AX? Why is it confirmatory?
Our Response: In the revised manuscript, we have now highlighted the reasons underlying each experimental decision in yellow.
ROS can cause double-strand breaks in the DNA. gH2AX foci formation is one of the earliest DNA damage response events and is therefore, used as confirmatory marker [1]. Hence, we used confocal microscope to look for the formation of gH2AX foci. However, as suggested by the reviewer in comment # 8, we have removed this data from the revised manuscript.
- The authors suggest in multiple places of the manuscript that they confirm their (RT-) qPCRs by Western blot. This is an absolutely wrong statement as mRNAs are not translated just because there is more of them. Multiple mechanisms operate over mRNA translation and stability including mRNAs that actually do different things than just being mere messengers; thus, the claim is not valid and must not be used.
Our Response: We agree with the reviewer that the western blot does not confirm the qRT-PCR data. We have removed this claim in the revised manuscript.
- In multiple places the authors provide statistical test’s information within the main text. This must be removed and placed in figure legends for each figure.
Our Response: As suggested, we have placed the statistical test’s information in the figure legend in the revised manuscript.
- Multiple data comparisons are scattered all over the figures and supplementary figures. I suggest the authors to establish valid comparisons within the same figures and to collect comparable data in single graphs rather than spanning them all across the manuscript.
Our Response: We have merged the results wherever it was possible without compromising on the figure legibility. For example, Fig. 2J and Fig.S3A.
However, it was not possible in case of the qRT-PCR data for the antioxidant genes. We have five antioxidant genes in each figure. If we merged two conditions in one single graph, the figure will be too large and become difficult to read. However, if the reviewer still feels that we should merge the figures, we are willing to do so.
- Many experiments add very little to the (I believe) overall message. For instance, the microscopy assay or the Western blot quantifications. All this should be either removed or send to supplementary.
Our Response: As suggested by the reviewers, we have now removed the microscopy assay from the manuscript. The western blot quantification is provided in the supplementary file.
- I suggest the authors to include the values of untreated/untransfected for every time point in a time course. Otherwise, the displayed data is not convincing.
Our Response: The time course experiment was done only with H2O2 treatment. (Fig. 1A). We have normalized the data with respect to the untreated samples. If the reviewer wants the raw data, we can provide that. We have not done any time course in case of transfection experiments.
The fold change values are also provided for clarification in the heat map Fig. S4.
- All the fragment regarding cell line specificity is rather mute. Should be removed as it does not bring anything to the table. Especially when only 2 cell lines are compared.
Our Response: As suggested by the reviewer, we have removed the data from the revised manuscript.
- Please make the figures auto-explicative by for instance replacing the word “treated” with H2O2.
Our Response: We have replaced “treated” with H2O2 in all the figures.
- How does the authors normalize their assays for transfection efficiency? Without this crucial piece of information, no conclusion can be made out of any of the transfection experiments.
Our Response: We have calculated transfection efficiency for each experiment and have mentioned it in the figure legends of the revised manuscript.
We have used scrambled shRNA as control for knockdown experiments. The empty vector (vector without any insert) has been used as control for the overexpression experiments.
However, we were unable to normalize for transfection between the control and the test group as we could not figure out a method to do so. We can use Renilla vector in case of luciferase assays but for knockdown experiments we were not sure what method to use.
- P-Values mean very little if no context is given for the statistical test. Is it an ANOVA, a two-tailed t-test, what is it? All this info must be included in the figure legend.
Our Response: Paired t-test was performed using Sigma plot software. We have now included this information in the revised manuscript.
- Antibody validation by intracellular localization as determined by microscopy is a no go.
Our Response: We have removed the localization data from the revised manuscript.
- The authors suggest that PICH regulates Nrf2 but the data displayed only supports a correlation. Please tone this down.
Our Response: We have reworded the sentences as suggested by the reviewer. It now reads as follows:
Taken together, the expression of PICH correlates with the expression of Nrf2 as well as the antioxidant genes both in the absence and presence of oxidative stress suggesting the protein might be regulating these genes. Finally, the ATPase activity of the protein appears to be required for transcriptional regulation as overexpression of the ATPase dead mutant failed to rescue the downregulation of genes in ShPICH cells.
- The wording is too strong in some places. For instance: Line 459… Nrf2 regulates the expression of PICH “only” on oxidative stress.
Our Response: We have reworded the sentence. It now reads as follows:
Thus, PICH expression correlated with Nrf2 expression both in the absence and presence of oxidative stress while Nrf2 expression appears to correlate with PICH expression during oxidative stress.
- Authors should show IgG instead of beads in their ChIP data for all ChIP data as the control values are not reported in half of the document.
Our Response: We did ChIP with IgG as well as with beads alone in case of untreated and treated cells (Figure enclosed for IgG experiments). As the data same pattern in both the cases, we used beads alone in all the subsequent experiments.
- Why the authors only measure 2 histone modifications??? They suggest that there may be 4 different histone modifications or even more by the restrict their analysis to only 2. Moreover, H3K27Ac is often found in enhancers not promoters albeit a few reports. This clearly needs to be discussed.
Our Response: The CBP/p300 histone acetyltransferase catalyzes H3K27 acetylation . Nrf2 has been shown to co-immunoprecipitate with p300 in HEK293T cells [2]. A crosstalk between H3K27ac and H3K4me3 has also been reported [3]. Therefore, we decided to investigate the role of these two modifications in our experiments. We have explained the rationale in the revised manuscript.
- The biochemical/structural analysis is completely out of place in here and the data is not sufficient to draw any conclusion. I suggest the authors to focus and what is to be the main message of the manuscript and display the data accordingly rather than tossing all together and trying to make it work.
Our Response: As suggested by the reviewer, we have removed the biochemical data in the revised manuscript.
- The discussion is way too long for the contents of the manuscript. The major and minor conclusions of the manuscript are not substantiated by the data and it is the opinion of this reviewer that the manuscript needs a major overhaul for it to work minimally as fundamental controls and experimental evidence are missing.
Our Response: We have shortened the discussion in the revised manuscript.
Minor comments:
- Please check manuscript wording and syntax for a future iteration.
Our Response: We have done a grammar check and have worked on the wording as well as syntax in the revised manuscript.
- All figures are displayed with low resolution. I suggest the authors to fix this. Place comparisons in single charts to enable direct comparison. I have the impression that all the data may be condensed in 2-3 figures. Please work on the logics all over the manuscript as the way the data is presented is not convincing at all.
Our Response: We have provided logic for every experiment. In the revised manuscript we have highlighted the logic in yellow.
We have provided high resolution figures in the revised manuscript.

Reviewer 3 Report
Dutta et al investigate the role of PICH in the oxidative stress response. PICH was found to activate transcription of NRF2 and other genes involved in the oxidative stress response. Furthermore, the depletion of PICH increased ROS and resulted in DNA damage. PICH was found to occupy promoters of antioxidant genes, which was associated with changes in DNA conformation.
Critique: The finding that PICH regulates antioxidant gene expression and the oxidative stress response is novel and important. The data are generally strong and support this conclusion. However, there is a major concern about the conclusion that the ATPase domain is required.
Major concern: The rescue experiments with PICHK128A in Figs. 2 and S2 are problematic. The expression of the plasmid is not the same as that of wildtype PICH in prior experiments, therefore, the claim that it does not rescue expression of NRF2 and other genes due to non-functional ATPase activity is an over-interpretation. In order to make conclusions about the ATPase domain, it will be important to get a higher expression level of mutant PICH in cells.
Other concerns:
1. Fig. 3H and I: The representative Western showing Nrf2 expression in Fig. 3H does not match the quantitation in Fig. 3I with respect to NRF2 expression in the rescue experiments. If NRF2 was really rescued as the quantitation indicates, there should be a better representative Western that should be shown. Also, it is not clear if the effect on PICH protein expression shown in Fig. 3L is significant.
2. The title is inappropriate since PICH does not have chromatin remodeling activity. Many manuscripts just state PICH in the titles without further characterization. If a description of PICH is to be included in the title, it would be more appropriate to refer to it as the PLK1-interacting checkpoint helicase.
3. Introduction: Lines 37-39, clarify that PICH does not have chromatin remodeling activity.
4. Results: Lines 274-279: Cite reference 21 for the DAAO plasmid. In Materials and Methods, indicate where this plasmid was obtained.
5. Some experiments do not indicate how many times they were performed:
Fig. S1B: How many independent experiments does the quantitation represent?
Fig. 1i: SMARCAL1 Western should be improved. It is not clear where the band is.
Fig. 1F and G: Indicate how many experiments the data represent.
Fig. 1L and Fig. S1G, Fig. 2E and S2E: Indicate how many catalase experiments, the data represent.
Figs. 2F-I, S2F-I, Fig. 2N and Fig. S2N: Indicate how many experiments, the data represent.
6. Fig. 4 B It is confusing that occupancy of PICH and NRF2 on the NRF2 promoter is higher in untreated samples and that NRF2 occupancy on the NRF2 promoter increases upon PICH knockdown (Fig. 6B). Discussion of this odd observation is needed with regard to NRF2 regulation.
Author Response
Dutta et al investigate the role of PICH in the oxidative stress response. PICH was found to activate transcription of NRF2 and other genes involved in the oxidative stress response. Furthermore, the depletion of PICH increased ROS and resulted in DNA damage. PICH was found to occupy promoters of antioxidant genes, which was associated with changes in DNA conformation.
Critique: The finding that PICH regulates antioxidant gene expression and the oxidative stress response is novel and important. The data are generally strong and support this conclusion. However, there is a major concern about the conclusion that the ATPase domain is required.
Major concern: The rescue experiments with PICHK128A in Figs. 2 and S2 are problematic. The expression of the plasmid is not the same as that of wildtype PICH in prior experiments, therefore, the claim that it does not rescue expression of NRF2 and other genes due to non-functional ATPase activity is an over-interpretation. In order to make conclusions about the ATPase domain, it will be important to get a higher expression level of mutant PICH in cells.
This experiment has been done multiple times. The mutant PICH is behaving as a dominant-negative here.
Our Response: We concur with the reviewer that the expression of the mutant is not the same as that of the wild type. We have removed this data from the revised manuscript.
Other concerns:
- 3H and I: The representative Western showing Nrf2 expression in Fig. 3H does not match the quantitation in Fig. 3I with respect to NRF2 expression in the rescue experiments. If NRF2 was really rescued as the quantitation indicates, there should be a better representative Western that should be shown. Also, it is not clear if the effect on PICH protein expression shown in Fig. 3L is significant.
Our Response: We have presented a better blot that matches the quantitation. The expression of PICH protein in the rescue experiments was same as that of the vector alone transfected sample. There was no significant difference in the expression of PICH between the control (vector alone) and the Nrf2 overexpression (in Nrf2 downregulated background).
In the revised manuscript, the quantitation is shown in Supplementary Figure 6 as the reviewer 2 wanted us to move all the western blot quantitation data to supplementary. Therefore, the quantitation of Fig. 3H is shown in Fig. S6M.
- The title is inappropriate since PICH does not have chromatin remodeling activity. Many manuscripts just state PICH in the titles without further characterization. If a description of PICH is to be included in the title, it would be more appropriate to refer to it as the PLK1-interacting checkpoint helicase.
Our Response: We have reworded the title:
PLK-1 interacting checkpoint helicase, PICH, mediates cellular oxidative stress response
- Introduction: Lines 37-39, clarify that PICH does not have chromatin remodeling activity.
Our Response: As suggested by the reviewer, we have clarified that PICH does not have chromatin remodeling activity. It now reads as follows:
PICH (PLK-1 interacting checkpoint helicase), also known as ERCC6L, is a Rad54-like helicase belonging to the ATP-dependent chromatin remodeling protein family [2–4]; however it does not have any chromatin remodeling activity
- Results: Lines 274-279: Cite reference 21 for the DAAO plasmid. In Materials and Methods, indicate where this plasmid was obtained.
Our Response: We have cited the reference for the DAAO plasmid and indicated in Materials and methods as to where we obtained this plasmid in the revised manuscript. The reference number is 23 in the revised manuscript.
DAAO plasmid [23] was a kind gift from Prof. S.K. Goswami, School of Life sciences, JNU.
- Some experiments do not indicate how many times they were performed:
Fig. S1B: How many independent experiments does the quantitation represent?
Each of the experiments is represented performed three individual times. Anything that differs from this is mentioned in the main text.
Fig. 1i: SMARCAL1 Western should be improved. It is not clear where the band is.
Our Response: All the experiments were performed as biological triplicate with two technical replicates for each biological experiment. For SMARCAL1, we have provided a better blot.
Fig. 1F and G: Indicate how many experiments the data represent.
Our Response: The experiment was done in triplicates and each time a minimum of 100 cells were taken for analysis. However, this data has been removed on the suggestion of Reviewer # 2.
Fig. 1L and Fig. S1G, Fig. 2E and S2E: Indicate how many catalase experiments, the data represent.
Our Response: The catalase activity experiments represented in the graph are the average of three individual experiments. This is now mentioned in the figure legend.
Figs. 2F-I, S2F-I, Fig. 2N and Fig. S2N: Indicate how many experiments, the data represent.
Our Response: The data represented are the average of three individual experiments and the in case of DCFDA assay, a minimum of 100 cells were taken into consideration while doing the analysis.
Fig. 4 B It is confusing that occupancy of PICH and NRF2 on the NRF2 promoter is higher in untreated samples and that NRF2 occupancy on the NRF2 promoter increases upon PICH knockdown (Fig. 6B). Discussion of this odd observation is needed with regard to NRF2 regulation.
Our Response: It is possible that Nrf2 is a negative regulator of its expression. However, additional experiments are needed to confirm this point. We have now mentioned it in the text.
Reviewer 4 Report
In this paper, the authors focus on the role of PICH and Nrf2 in the oxidative stress response. While Nrf2 was known to be involved in this process, PICH appears to be a novel component, functioning as a co-regulator with Nrf2. The data presented is convincing and consistent with this conclusion.
The authors highlight the interesting difference between the endogenous and exogenous oxidative pathways, which differ in which genes are upregulated. This is indeed interesting and I would suggest to incorporate some of Fig S1 into Fig 1, to clearly contrast how these treatments differ in their effect on BRG1 and SMARCAL1.
One of the weak points of the paper is the introduction, which presents a brief overview of some roles of PICH and Nrf2, but it is unclear why the authors decided to focus on these two proteins and on the oxidative response. How were these proteins identified to be candidates for this study? The second paragraph focuses on PICH’s role at kinetochores – what is the scientific basis that made the authors decide to check whether this kinetochore-associated protein played a role in the oxidative stress response? Why were PICH and Nrf2 assayed in the D-serine experiment upon seeing that neither BRG1 nor SMACAL1 changed expression? (line 285) – The answer to this should probably constitute a good starting point, but the introduction should probably be entirely re-written.
Another weak point is that there is no justification for using SEM (apart from decreasing error bars); in general, barplots should not display SEM, but SD.
Fig 1A: it is quite surprising that the levels of PICH, while up at 20 and 30 mins, are very strongly down at 40mins then very high at 50mins; this may indicate that more biological replicates may be necessary.
Fig 1D: a scale bar needs to be added. The scale bar in Fig 1F is not legible, and should be corrected.
Fig 7A: the line point to residue 128 points to T rather than K
Fig 7 would be made much clearer by adding a representation of the different templates used in panels C and D, as well as the concentration of protein used.
Figure 8 has no legend. The difference between the models is very small, and could be better summarized in a single model with a well-placed interrogation mark (?) to indicate that it is still unclear whether Nrf2 or PICH binds to DNA directly and in which order.
Furthermore, a lot of formatting needs to be double-checked, as there are inconsistencies; for example, some fonts are not identical in the text; some other parts show highlights (Figure 1 legend); furthermore, it seems like lines 318-325 were meant to be part of the legend of Figure 1 rather than part of the Results section. Same for lines 395-401 and Figure 2. Same for lines 482-486 and Figure 3.
Author Response
Comments and Suggestions for Authors
In this paper, the authors focus on the role of PICH and Nrf2 in the oxidative stress response. While Nrf2 was known to be involved in this process, PICH appears to be a novel component, functioning as a co-regulator with Nrf2. The data presented is convincing and consistent with this conclusion.
The authors highlight the interesting difference between the endogenous and exogenous oxidative pathways, which differ in which genes are upregulated. This is indeed interesting and I would suggest to incorporate some of Fig S1 into Fig 1, to clearly contrast how these treatments differ in their effect on BRG1 and SMARCAL1.
- One of the weak points of the paper is the introduction, which presents a brief overview of some roles of PICH and Nrf2, but it is unclear why the authors decided to focus on these two proteins and on the oxidative response. How were these proteins identified to be candidates for this study? The second paragraph focuses on PICH’s role at kinetochores – what is the scientific basis that made the authors decide to check whether this kinetochore-associated protein played a role in the oxidative stress response? Why were PICH and Nrf2 assayed in the D-serine experiment upon seeing that neither BRG1 nor SMACAL1 changed expression? (line 285) – The answer to this should probably constitute a good starting point, but the introduction should probably be entirely re-written.
Our Response: We have provided an explanation for investigating the role of PICH in oxidative stress in the Introduction:
Previously, we had shown that the expression of BRG1 and SMARCAL1, two ATP-dependent chromatin remodelers, was upregulated both at the transcription and protein level in doxorubicin-treated HeLa cells [5]. We showed that this upregulation was needed for the activation of the DNA damage response pathway [6,7]. To understand whether this upregulation was needed during oxidative stress, we performed a preliminary experiment wherein we transfected HeLa cells with D-amino acid oxidase gene. On treatment with D-serine, H2O2 was produced inside the cells resulting in oxidative stress. We then analyzed expression of ATP-dependent remodelers that are known to play a role in DNA damage response/repair. In this experiment, we used PICH as a negative control as there were reports stating that this protein cannot remodel nucleosomes. Further, there are no reports that PICH plays a role in DNA damage response/repair. To our surprise, we found that BRG1, SMARCAL1, RAD54L, ZRANB3, and INO80 were unchanged but PICH was upregulated. This led us to hypothesize that maybe PICH has a role to play in oxidative stress either in DNA damage response pathway or in modulating expression of antioxidant genes as well as of Nrf2.
In the manuscript, we have provided data only for BRG1, SMARCAL1 and PICH. We are providing the entire dataset for reviewer’s perusal:
- Another weak point is that there is no justification for using SEM (apart from decreasing error bars); in general, barplots should not display SEM, but SD.
Our Response: As we have averaged data from biological replicates, we have used SEM instead SD. We also referred some papers to see how the data has been represented in such cases before making the decision [8–10]. We have written this justification in the Materials and methods section of the revised manuscript:
As we have analyzed data from biological replicates, we have calculated SEM instead of standard deviation (SD).
In case of biochemical data, we did use SD. However, that data has been now removed on the recommendation of reviewer# 2.
- Fig 1A: it is quite surprising that the levels of PICH, while up at 20 and 30 mins, are very strongly down at 40mins then very high at 50mins; this may indicate that more biological replicates may be necessary.
Our Response: Three biological replicates were performed and the data is presented as an average of these biological replicates. We agree that there is a difference between the transcript levels and the protein levels. It is quite possible that while the transcript levels fluctuate as a response to oxidative stress, the protein levels remain steady. Thus, the mRNA levels might be transcriptionally regulated while post-transcriptional mechanisms might be ensuring that the protein expression does not change. This is a point to be investigated in future. We have provided the following explanation in the text of the revised manuscript:
It needs to be noted that the protein levels remain upregulated steadily post-H2O2 treatment while the RNA levels appear to fluctuate. It is quite possible that the expression of PICH is regulated by both transcriptional and post-transcriptional mechanisms ensuring that protein levels are steady in the cell even if the transcript levels fluctuate.
- Fig 1D: a scale bar needs to be added. The scale bar in Fig 1F is not legible, and should be corrected.
Our Response: We apologize for the error. We have rectified them in the revised manuscript.
Fig 7A: the line point to residue 128 points to T rather than K
Our Response: We have removed the biochemical data from the revised manuscript as suggested by Reviewer 2.
Fig 7 would be made much clearer by adding a representation of the different templates used in panels C and D, as well as the concentration of protein used.
Our Response: As suggested by the Reviewer # 2, this data has been now removed from the revised manuscript.
Figure 8 has no legend. The difference between the models is very small, and could be better summarized in a single model with a well-placed interrogation mark (?) to indicate that it is still unclear whether Nrf2 or PICH binds to DNA directly and in which order.
Our Response: As suggested by the reviewer, we have merged the model and add interrogation marks. We also have added the figure legend in the revised manuscript, and it is Figure 7 now.
Furthermore, a lot of formatting needs to be double-checked, as there are inconsistencies; for example, some fonts are not identical in the text; some other parts show highlights (Figure 1 legend); furthermore, it seems like lines 318-325 were meant to be part of the legend of Figure 1 rather than part of the Results section. Same for lines 395-401 and Figure 2. Same for lines 482-486 and Figure 3.
Our Response: We have corrected all the formatting errors in the revised document.
Round 2
Reviewer 3 Report
The paper is much improved. I have one comment. The third line of the Abstract still indicates that PICH is an ATP-dependent chromatin remodeler. This is not accurate and should be changed to "a RAD-54-like helicase" belonging to the ATP-dependent chromatin remodeling protein family."
Author Response
Response to the Reviewer:
Comments and Suggestions for Authors:
The paper is much improved. I have one comment. The third line of the Abstract still indicates that PICH is an ATP-dependent chromatin remodeler. This is not accurate and should be changed to "a RAD-54-like helicase" belonging to the ATP-dependent chromatin remodeling protein family."
Our Response: We thank the reviewer for the kind comments. We have modified the abstract as suggested by the reviewer. The line now reads:
The study presented here shows that the expression of PICH, a Rad54-like helicase belonging to the ATP-dependent chromatin remodeling protein family, is upregulated during oxidative stress in HeLa cells.